# Chemical and Synthetic Biology Approaches for Cancer Vaccine Development

**DOI:** 10.3390/molecules27206933

**Published:** 2022-10-16

**Authors:** Farzana Hossain, Shruthi Kandalai, Xiaozhuang Zhou, Nan Zhang, Qingfei Zheng

**Affiliations:** 1Department of Radiation Oncology, College of Medicine, The Ohio State University, Columbus, OH 43210, USA; 2Center for Cancer Metabolism, James Comprehensive Cancer Center, The Ohio State University, Columbus, OH 43210, USA; 3Department of Biological Chemistry and Pharmacology, College of Medicine, The Ohio State University, Columbus, OH 43210, USA

**Keywords:** cancer vaccine, tumor antigen, peptide epitope, bioconjugation, vaccine delivery system

## Abstract

Cancer vaccines have been considered promising therapeutic strategies and are often constructed from whole cells, attenuated pathogens, carbohydrates, peptides, nucleic acids, etc. However, the use of whole organisms or pathogens can elicit unwanted immune responses arising from unforeseen reactions to the vaccine components. On the other hand, synthetic vaccines, which contain antigens that are conjugated, often with carrier proteins, can overcome these issues. Therefore, in this review we have highlighted the synthetic approaches and discussed several bioconjugation strategies for developing antigen-based cancer vaccines. In addition, the major synthetic biology approaches that were used to develop genetically modified cancer vaccines and their progress in clinical research are summarized here. Furthermore, to boost the immune responses of any vaccines, the addition of suitable adjuvants and a proper delivery system are essential. Hence, this review also mentions the synthesis of adjuvants and utilization of biomaterial scaffolds, which may facilitate the design of future cancer vaccines.

## 1. Introduction

Cancer vaccines, an essential part of immunotherapy, aim to induce long-lasting antigen-specific immunity to eliminate tumor cells and prevent a recurrence. Historically, many cancer vaccines have been prepared from tumor cells, immune cells, and live attenuated or inactivated bacteria and viruses [1]. However, drawbacks, such as severe infections, pathogen mutations, and large-scale cultures of pathogenic organisms, and potentially unforeseen reactions have led to the construction of synthetic tumor vaccines [1]. In recent years, many chemical and synthetic biology approaches have been developed to target novel tumor biomarkers and these have made significant progress in eliciting stronger immune responses. Cancer cells of diverse origins overexpress tumor-specific antigens (TSAs) or tumor-associated antigens (TAAs) on their cell surfaces, due to aberrant glycosylation. These have been long considered biomarkers for cancer detection and have been prioritized as tumor antigens (TAs) for vaccine development [2]. Furthermore, specific mutations occurring in tumor cells from cancer patients may generate novel epitopes of self-antigens, which are referred to as neo-antigens or neo-epitopes. Recent scientific advances have enabled the identification of tumor-specific mutations and the development of personalized therapeutic cancer vaccines that are customized to target tumor rather than normal cells of individual patients, thereby significantly facilitating targeted cancer therapies [3]. Ususlly, the tumor antigens contain extracellular oligosaccharides, which are linked to proteins or lipids through the involvement of different glycosyltransferases or mutation events [2]. Therefore, the antigens or epitopes as crucial components of cancer vaccines, are generally small sequences of carbohydrates or amino acids that can be chemically synthesized via glycosylation, peptide synthesis, or chemoenzymatically from isolates [4]. Furthermore, one-pot synthesis and solid-phase synthetic chemical strategies provide the foundation for rapid preparation of antigens, thereby allowing for the development of multicomponent vaccines.

In general, for a cancer vaccine to be effective, the desired antigens need to be delivered to antigen-presenting cells (APCs), which most notably include dendritic cells (DCs), as well as macrophages, neutrophils, and lymphatic endothelial cells. Subsequently, the APCs should process and cross-present TAs and activate T-cells (naïve CD4^+^ T cells and CD8^+^ T cells) in lymph nodes [5]. Lastly, activated T-helper cells (T_h_-cells) further activate B-cells, which differentiate into clones of plasma and memory B-cells and produce antibodies against the antigens (Figure 1) [2,6]. Additionally, antibody-dependent cellular cytotoxicity (ADCC) can occur when generated antibodies bind with antigen-presenting tumor cells and are recognized by Fc receptors on natural killer (NK) cells, leading to the release of granzymes [2]. However, these small molecular antigens are B-cell dependent, poorly immunogenic on their own, and therefore need to be conjugated to carrier proteins to stimulate or boost an immune response (Figure 2) [4]. Due to this limitation, many researchers began to conjugate tumor-associated carbohydrate antigens (TACAs) with T-cell stimulating protein carriers, by using connecting linkers [7]. Initially, the responses of these vaccines were promising, but further studies reported that these protein carriers can be self-immunogenic and suppress antigen-specific immunogenicity [8]. Subsequently, TACAs have been also coupled with zwitterionic polysaccharides (ZPs) [9] and Toll-like receptor (TLR) ligands [10], among others, to develop semi-synthetic to fully synthetic self-adjuvating cancer vaccines. This review will highlight the different bioconjugation approaches, such as the use of biorthogonal or biocompatible linkers, carriers, and click chemistry, which have been utilized to develop TACA-based-conjugate vaccines. To boost immune response, recent vaccines have been formulated by combining with synthetic adjuvants. Therefore, highly potent agonist screening and synthesis need to be addressed with chemical strategies that hold the most promise for enhanced immunotherapy [1].

Modulation of biological systems through synthetic biology is an emerging discipline that focuses on the predictable reprogramming of living cells [11,12]. Major focuses of synthetic biology in vaccine development include the development of new genetic parts from nucleic acids and synthesis of antigens that are then transfected into cells or pathogens, biomaterial-based delivery systems, and synthetic receptors [13]. Many synthetic biology approaches can be utilized to alter the direct use of live pathogens in cell-based vaccines, especially via genetic modification of cells and the use of dead tumor cells, engineered bacteria, plant-based antigens, synthetic DNAs or messenger RNA (mRNAs), and inactivated pathogens, toxins, and virus-like particles (VLPs) [1,12,13]. Genetically engineered cancer vaccines can deliver a plethora of known and unknown antigens to the host immune system through epitope spreading, and therefore can be utilized for development of personalized tumor vaccines [14]. In this review, we have discussed the basic design conditions and clinical status of several classes of cancer vaccines (e.g., carbohydrate, nucleic acid, peptide/protein, cell-based) (Table 1), along with novel biomaterial-based delivery platforms that improved their safety and efficacy. Among different kinds of cancer vaccines, genetically modified cancer vaccines often use recombinant vectors, derived from live cells or pathogens [13,15]. We have also discussed the transfection of chemokines, cytokines, mRNA, TAAs, TSAs, and T-cell costimulatory molecules into tumor cells, DCs, viruses, bacteria, and plants via different gene modification strategies, which can attract synthetic biologists to design improved tumor vaccines [13,16].

Even though many chemical and synthetic biology approaches have been developed and applied to treat a variety of cancer types, only a few cancer vaccines have been approved by the U.S. Food and Drug Administration (FDA) over the past few decades [including sipuleucel-T (provenge^®^), T-VEC (Imlygic^®^), cervarix^®^, and gardasil^®^]. A major limitation of cancer vaccines could be inefficient delivery in vivo, where administered vaccines cannot successfully reach their desired targets [15]. Therefore, cancer scientists are focusing on developing new delivery materials for the next generation cancer vaccines. Various delivery approaches, such as nanoparticles, hydrogels, self-assembled peptides or proteins, and other biomaterial scaffolds, have been widely utilized in combination with various forms of cancer vaccines and their preclinical outcomes are promising [5,19,20].

## 2. Chemical Approaches for the Construction of Tumor Antigens, Vaccines, and Improving Potency

### 2.1. Carbohydrate Antigens

Cancer cells of diverse origins overexpress different extracellular TACAs compared to normal cells, due to aberrant glycosylation mediated by different glycosidases [2]. Because of this distinct characteristic, TACAs have always been prioritized as antigens while designing pre-clinical therapeutic cancer vaccines. These carbohydrate antigens are anchored into the lipid bilayers [e.g., the glycolipids Globo-H, GM_2_, GD_2_, and GD_3_] or are part of proteins [e.g., glycoproteins that have an *N*-acetylgalactosamine core (GalNAc) oligosaccharide, such as T_N_, T_F_, sialyl-T_N_ (ST_N_), and MUC1] [7,21]. However, the difficulties associated with isolating TACAs through cancer cell culture have prompted the rise of chemical synthesis as their primary means of access. Therefore, various synthetic strategies have been developed to prepare these complex oligosaccharide antigens.

#### 2.1.1. Glycosylation/Glycal Assembly

Some antigenic glycans are expressed in low amounts in bacterial cultures; therefore, chemical synthesis can provide a larger yield of carbohydrate antigens for vaccine preparation [1]. The formation of the glycosidic bond via glycosylation is the primary means by which different monosaccharide building blocks are assembled into more complex oligosaccharide structures. However, in this strategy, the unique challenge is in the regiochemical and stereochemical selective control [22]. The carbohydrate coupling partner, which contributes its anomeric carbon to the anomeric linkage, is considered a glycosyl donor and acts as an electrophile. The corresponding coupling partner, considered as a glycosyl acceptor, acts as the nucleophilic counterpart. This strategy typically relies on a selectively protected glycosyl donor, which incorporates a suitable leaving group at its anomeric center. A suitable electrophilic ‘activator’ [e.g., trifluoromethane sulphonic acid (TfOH), trimethylsilyl trifluoromethanesulfonate (TMSOTf), silver triflate (AgOTf), indium (III) bromide (InBr_3_), I_2_, or *N*-Iodosuccinimide (NIS)] is chosen depending on the anomeric leaving group [e.g., thio-phenyl (SPh), thio-ethyl (SEt), halides, trichloroacetimidates (TCA), acetyl (OAc), *p*-methoxybentyl ether (PMB), and tolylthio (STol)] [23,24,25]. The removal of leaving group results in the formation of an oxocarbenium ion, which makes the anomeric functionality electron-deficient, thereby allowing anomeric substitution by the nucleophilic glycosyl acceptor (Nu-OH) to form a desired glycoconjugate. The alpha (*α*) and beta (*β*) selectivity are contingent upon several conditions, including temperature, neighboring group participation, and time. Following the glycosylation procedures, multiple coupling steps have been successfully performed by many laboratories to generate various lengths of complex oligosaccharide antigens.

Furthermore, to facilitate the rapid synthesis of oligosaccharides, the “one-pot sequential glycosylation strategy” [26] has been getting much attention, as it can synthesize many carbohydrate antigens at once (Figure 3). This strategy relies on the different reactivity of similar (chemo-selective) or chemically diverse (chemo-orthogonal) anomeric leaving groups. This process is also based on the observation that a significant difference between the reactivities of different glycosyl donors can be achieved by varying the electron-donating or electron-withdrawing characters of the anomeric leaving groups and other protecting groups [26]. Some donors contain both protected and unprotected hydroxyl groups based on the desired glycosylation position. Wong et al. synthesized Globo-H following this one-pot synthetic strategy [27]. A similar one-pot multiple glycosylation process was applied to a mixture of the trisaccharide, a fucosyl donor, and the lactose-derived acceptor, assuring synthesis of the Globo-H hexasaccharide [28]. The most significant advantage of this one-pot strategy is the assembly of oligosaccharide cores with minimal isolation and purification of reaction intermediates at each step. [28] However, before conjugating any carbohydrate antigens to a carrier protein or lipid, a global deprotection protocol is typically applied to deprotect all the protecting groups [acetyl (Ac), benzoyl (Bz), levulinyl (Lev), benzyl (Bn), etc.] [25], followed by connecting these with a suitable linker. Other synthetic strategies, such as solid phase automated synthesis to assemble suitably functionalized glycal monomers sequentially to a solid support to prepare complex oligosaccharides, have enabled faster vaccine candidate preparation.

#### 2.1.2. Chemoenzymatic Synthesis

Aberrant expression of gangliosides, sialic acid-containing glycosphingolipids, has been broadly connected with cancer progression [2,29]. Therefore, gangliosides (e.g., GD_2_, GD_3_, fucosyl GM_1_, and GM_3_) are attractive TACA for anti-cancer vaccine development, but have been difficult to synthesize. To overcome challenges, including regioselectivity and stereoselectivity in sialylation and lower yields while glycosylating the oligosaccharide with the lipid, many groups have reported the chemoenzymatic synthesis of those gangliosides [30]. Chen et al. reported chemoenzymatic strategies for synthesizing common ganglioside-related cancer antigens, including GM_3_, the simplest ganglioside often utilized to form more complex gangliosides [31]. The biosynthesis of the corresponding GM_3_ sphingosine (GM_3_*β*Sph) lacking a fatty acyl chain was initiated from Lac*β*Sph and a monosaccharide *N*-acetylneuraminic acid (Neu5Ac) using a one-pot two-enzyme sialylation system containing *Neisseria meningitidis* derived CMP-sialic acid synthetase (NmCSS) and *Pasteurella multocida* derived *α*2-3-sialyltransferase 3 (PmST_3_) [31,32] where the Lac*β*Sph was previously prepared from lactose and phytosphingosine on a large scale [31]. For purification, a simple C18-cartridge SPE was used, which was able to remove excess Neu5Ac, cytidine 5′-monophosphate (CMP)-Neu5Ac, cytidine5′-triphosphate (CTP), and other byproducts. Similarly, other gangliosides were synthesized following this sequential one-pot multienzyme (OPME) glycosylation system [31].

### 2.2. Carbohydrate-Based Vaccines

As mentioned above, TACAs are significant biomarkers for developing therapeutic and preventive vaccines because they are prominent on tumor cell surfaces [2]. Furthermore, they are B cell-dependent, poorly immunogenic, and unable to induce cellular T cell-mediated CD4^+^ and CD8^+^ dependent immunity on their own, which is critical for the presentation of major histocompatibility complex (MHC) molecules on APCs [24]. Therefore, to stimulate T-cell responses, early attempts were focused on developing carbohydrate-protein conjugate vaccines by conjugating synthetic TACAs with a carrier protein (Figure 2), such as keyhole limpet hemocyanin (KLH), bovine serum albumin (BSA), diphtheria toxoid (DT), tetanus toxoid (TT), human serum albumin (HSA), ovalbumin, meningococcal outer membrane protein complex (OMPC), *Hemophilus influenzae* protein D, *Pseudomonas aeruginosa* exotoxin A (rEPA), etc. [24,33]. Here, the glycoprotein KLH, a biological product derived from giant keyhole limpets, is commonly used in pre-clinical studies, and DT is a formaldehyde-inactivated diphtheria toxoid [1]. Another formaldehyde-inactivated neurotoxin, TT, is widely used as a carrier in vaccines as it is readily available from *Clostridium tetani* cultures and its safety has been well-established [1]. Nowadays, a cross-reacting material 197 (CRM197), arising out of a single amino acid mutation, has been widely employed in clinical vaccines due to its low toxicity and the large-scale production from the *Corynebacterium diphtheria* C7(β197)^tox–^ strain to provide a high purity protein [1,34].

Over the past few years, many research teams have contributed enormously to carbohydrate-based vaccine development research. For example, Livingston-Danishefsky et al. reported the synthesis of several TACAs, including Globo-H, Lewis^y^, Lewis^x^, Lewis^b^, KH-1, MUC1, GM2, STn, and Tn [35]. They preclinically evaluated monovalent conjugate vaccines, as well as some multicomponent vaccines containing different oligosaccharide (TACAs) glycoconjugates on a polypeptide backbone, before finally linking it to KLH (Figure 4a) [35,36]. Furthermore, multiple protein-conjugated vaccines have reached randomized phase III clinical trials, including some KLH conjugates, such as Theratope, OPT-822 (Globo H-KLH), and GM2-KLH [33]. However, Theratope (sTn-KLH) conjugate formulated with an adjuvant (QS-21) was tested in a phase III trial on more than 1000 breast cancer patients and was proven to be safe, though no improvement in survival rate was observed [33]. Furthermore, Wong et al. have synthesized Globo-H vaccines using several protein carriers, including KLH, DT, TT, and BSA [37]. However, most carrier proteins, especially KLH and BSA, are auto-immunogenic and can suppress antibody production against several antigens [8]. Due to this limitation, many research groups are focusing on fully synthetic multicomponent vaccine constructs with specific T_h_-cell glycopeptide epitopes as immunostimulant adjuvants.

### Bioconjugation

The conventional method for conjugating any antigens with a carrier protein involves the utilization of a linker moiety. Several linkers like succinimide esters, *N*-succinimidyl 3-(2-pyridyldithio)propionate, *m*-maleimidobenzoyl hydrazide, 3-(bromoacetamido)-propionate, squaric acid diesters, 4-(4-*N*-maleimidomethyl)cyclohexane-1-carboxyl hydrazide, and *p*-nitrophenol esters, have been utilized depending on the compatible functional groups on the TACAs and carrier protein [38]. However, many groups are investigating if immunogenicity of the desired TACA or epitope in protein conjugate-based vaccine is being suppressed by the self-immunogenic protein carrier or even by the linker [6]. Other concerns regarding linkers, such as chain length, solubility issues, low yield, and first attachment to either protein or antigen, are pushing researchers to explore the alternative of carrier proteins with linker-free bioconjugation [2]. It has been reported that, like some carrier proteins, some ZPs are also known to elicit a CD4+ T-cell-dependent immune response, invoke class switching from IgM to IgG, and bind with Toll-like receptor-2 (TLR2) on DCs, which plays an active role in releasing interleukin 12 (IL-12) and interferon gamma (IFN-γ) [39,40]. Therefore, these polysaccharides encapsulating pathogenic bacteria have been a target for conjugate cancer vaccine construction for years [40]. Considering the previously stated concerns related to proteins and linkers, Andreana et al. has synthesized aldehyde reactive TACAs, like aminooxy-Tn, TF, STn, and Globo-H antigens, and conjugated them with several oxidized ZPs, aiming to develop linker-free and entirely carbohydrate-based semi-synthetic cancer vaccines (Figure 4b) [9,24,41]. As an alternative to carrier proteins, ZPs, like PS A1 and PS B, were isolated from anaerobic *Bacteroides fragilis* (ATCC 25285/NCTC 9343) and specific type 1 polysaccharide (Sp1) isolated from *Streptococcus pneumoniae* serotype 1 [24]. The vicinal *cis*-diol of the glucofuranose unit of PS A1 was oxidized to an aldehyde with sodium metaperiodate (NaIO_4_) in an acetate buffer [24]. Next, an oxime bond was formed between TACA-ONH_2_ and freshly oxidized PS A1. The vaccine candidate was then dialyzed to eliminate the buffer, followed by lyophilization.

### 2.3. Protein/Peptide-Based Epitopes and Vaccines

When cells begin to mutate into tumors, they start producing mutated proteins not found in healthy cells. These faulty proteins, also called neoantigens, can alert the body’s immune system, triggering T-cells that recognize those neoantigens to start destroying the cancerous cells [1,17,42]. Therefore, protein and peptide-based therapeutic cancer vaccines have attracted enormous attention in recent years in cancer-related research [17]. The major challenge while developing an effective vaccine against a specific protein is selecting an immunogenic peptide epitope from the target sequence for conjugation to a carrier [1]. Furthermore, the chosen epitope sequence should be unique to a particular protein to avoid or limit unwanted cross-reaction of antibodies generated by other similar proteins [43]. In addition, the peptide epitope sequence should be selected based on the possibility of the motif binding with MHC molecules displayed on the surface of APCs [1]. Many bioinformatic tools and analysis methods are available to rapidly identify the desired sequence from new target proteins, neo-epitopes or bacterial isolates [44].

Most peptide-based vaccines contain distinct 8–12 amino acids (aa) as epitopes from TA coding sequences, stimulating CD8^+^ T-cells or CD4^+^ T-helper cells against those antigens [45]. Predominantly, TAs are proteins or glycopeptides that are overexpressed in different carcinomas. They can be internalized into DCs, before being degraded into peptides and assembled into human leucocyte antigen (HLA) molecules on DCs surface for T-cell activation [2]. HLA is the expression product of the human MHC-complex, which is related to T-cell and B-cell activation followed by antibody production [2,10]. Therefore, the demonstration of cellular and humoral immune responses is essential for a vaccine-in-development to be considered potent. An example of such a TA is transmembrane glycoprotein MUC1 (mucin-1), which is regarded as a potential target for vaccine construction, and whose specific peptide sequences (SAPDTRPA) have been reported to be responsible for cytotoxic T-cell activation via formation of MHC I-complex [46]. Usually, the peptide backbone of MUC1 consists of a variable number of 20 amino acids with the repeating sequence of GVTSAPDTRPAPGSTAPPAH, which also contains glycosylation sites to form specific TAs [47]. A variety of new MUC1 conjugate synthetic vaccines and their immune responses have been reported, following conjugation with different protein carriers like KLH, TT, BSA, etc. [47]. For example, multicomponent fully synthetic MUC1 vaccines were constructed by Boons et al., conjugating the variable number of tandem repeat (VNTR) units, including different TACAs in the presence of varying TLR agonists (Pam_3_CysSK_4_) [48].

Among the numerous T-epitopes, a poliovirus-derived 13 amino acid peptide sequence KLFAVWKITYKDT (103-115, PV peptide) is one of the promising candidates to efficiently bind with mouse MHC class II molecules and elicit T-helper cell-dependent immune responses [49]. This peptide sequence was derived from capsid VP1 proteins of poliovirus type 1 (PV1 Mahoney) [49]. This Mahoney strain and VP1 capsid protein also generated MHC I mediated cytotoxic T-cell activation responses in mice [50]. Along these lines, two VP1 peptide sequences, (110-120, TYKDTVQLRR) and (202-221), were revealed that contain an MHC I K^d^ binding motif. On the other hand, survivin, an anti-apoptotic protein, has also been a peptide-based cancer vaccine design target. A survivin-based vaccine containing a pool of three synthetic long peptides (SLPs) with eight CD4^+^ epitopes and six CD8^+^ epitopes was developed by Tanchot et al [51].

#### 2.3.1. Peptide Synthesis

Nowadays, many cancer vaccines contain specific sequences of peptide epitopes as the backbone for connecting with TACAs. Most of the iterative synthesis of long-chain peptides has become automated and widespread, which is much faster than traditional solution phase synthesis. Solid-phase peptide synthesis protocols are utilized to expedite synthesis, increase yield, and avoid tedious purification while synthesizing the extended chain of peptide epitopes. Different polymeric resins, such as trityl, 2- chlorotrityl resin, PAM resin, Wang resin, and MBHA (methylbenzhydryl amine) are used as solid support [48,52,53]. The side chains of the amino acids are protected by some base-sensitive groups, especially Fmoc or acid-sensitive groups (e.g., Boc, ^t^Bu, Trt, and Pbf). Some activating reagents (e.g., EDCl, HBTU/HBOt, HATU/HAOt, and PyBOP), have been utilized for amide bond formation [53,54]. The commonly used bases for Fmoc deprotection are weak bases, including piperidine and morpholine, as any strong bases (NaOH or LiOH) are avoided because of possible racemization [55]. If the carboxylic group is given allyl protection, Pd-based catalysts are used during deprotection [54], whereas if benzyl (Bn) group was used for protection, a hydrogen atmosphere is applied in the presence of Pd/C [56]. Usually, the glycopeptide building blocks are synthesized separately following traditional glycosylation procedures, then introduced into the peptide backbone, followed by solid-phase synthesis via amide bond formation. After completion of the synthesis, peptide epitopes are cleaved off from the solid resin under an acidic condition (TFA or TIPS/TIPS/H_2_O) and are purified by HPLC on a C18 reverse-phase column [48,53,57]. The desired epitope is utilized for the vaccine construction after purification. For example, Boons et al. synthesized a Tn-MUC1-based multicomponent cancer vaccine, where the glycosylated amino acid *N*-Fmoc-Thr-(AcO_3_-α-D-GalNAc) was introduced manually under microwave heating with the APGSTAPPAHGVTSAPDTRPAP peptide chain [48]. Furthermore, as illustrated in Figure 4c, the cyclo-decapeptide scaffold of a self-adjuvating multivalent glycolipid-peptide (GLP) cancer vaccine was synthesized based on the TASP (template assembled synthetic protein) model [58]. These regioselectivity addressable functionalized template (RAFT) molecules allow sequential and regioselective assembly of biomolecule-based ligands and biologically functional units to grant immune recognition and prevent steric hindrance [58].

#### 2.3.2. Bioconjugation

Peptide or glycopeptide epitope-based bioconjugation strategies often uses amide-bond-forming reactions between the lysine residues on the carrier protein surface and a carboxylic acid in the linker-containing epitope [1]. Alternatively, a bifunctional linker containing acid or amine can be used for coupling to the epitope and a maleimide or thalidomide group (which can be converted to an aminooxy group) to enable subsequent coupling to a modified carrier protein [1]. However, with the aim of targeted delivery of antigens, Sucheck et al. conjugated murine IgG3-Fc with a MUC1-containing cancer vaccine (Figure 4d) [59]. Initially, the Fc portion of the murine IgG3 isotype was incorporated into a heterobifunctional cross-linker *N*-succinimidyl 3-(2-pyridyldithio)propionate (SPDP) [59]. Then, the disulfide bond was reduced into a thiol residue using dithiothreitol (DTT) and the freshly prepared thiolated Fc solution was conjugated to dipalmitidoylphosphatidylcholine (DPPC) liposomes containing 1,2-distearoyl-*sn*-glycero-3-phosphoethanolamine-*N*- [maleimide(polyethyleneglycol)] DSPE-PEG-MAL [59]. Another example of peptide or protein-based bioconjugation is the synthesis of a three-in-one protein conjugate TLR7a-BSA-MUC1 as shown in Figure 4e [60]. A small-molecule TLR7 agonist (TLR7a) was converted to active ester TLR7a-NHS for reaction with the BSA protein to form conjugates [60]. In addition, a Tn-MUC1 antigen [GVTSAPDT(Tn)RPAPG] was synthesized and conjugated to BSA through the squaric acid diethyl ester linker [60]. Then, the TLR7a was covalently attached to the MUC1-BSA conjugate. Mass spectrometric (MALDI-TOF) analysis indicated an average of 6–7 TLR7a and 9–11 MUC1 glycopeptides covalently linked to BSA [60].

#### 2.3.3. Click Chemistry

A complex structure might arise during the construction of a cancer vaccine via bioconjugation between multiple reactive amino acid functional groups in the peptide or glycopeptide epitope, which can further compete with the connecting linker, resulting in the poor presentation of the desired epitope to the immune system [1]. To sidestep that issue, biorthogonal click chemistry can be used as an alternative to carrier protein coupling, thereby creating a more reliable immune response. Traditionally, click chemistry includes the formation of a 1,2,3-triazole via the copper(I)-catalyzed azide-alkyne cycloaddition (CuAAC) that can be used to link an azido-functionalized peptide epitope to an alkyne-functionalized carrier [61]. For example, Sucheck et al. synthesized a cancer vaccine conjugate (Pam_3_CysSK_4_-DBCO-MUC1-VNTR-TACA) (Figure 4f) containing Pam_3_CysSK_4_, a TLR agonist, linked via copper-free cycloaddition chemistry to a glycopeptide derived from the tumor marker MUC-1 also holding the Tn-antigen [62]. Furthermore, Betrozi et al. constructed glyco-polypeptides on polystyrene beads via click chemistry to mimic natural branched ligands that were found to trigger various APC-mediated cellular immune responses [4,63]. This *N*-carboxyanhydride (NCA) driven polymerization strategy is highly chemically controllable and may provide a novel method for conjugating and clustering glycoside-based agonists [63]. A chemical scaffold was reported by Jones et al. for multiple epitope presentation, which uses two sequential and orthogonal click reactions, a strain-promoted azide-alkyne conjugation (SPAAC) and a CuAAC reaction, to attach two different azide-containing peptides to a carrier protein in a simple one pot process [64]. Therefore, this strategy can be suitable for constructing more controllable polyvalent cancer vaccines. More vaccine candidates are beginning to emerge that have utilized these as linkers. However, further studies are required to analyze the immunogenic potential of the triazole ring formed following these click reactions.

### 2.4. Immunostimulant Adjuvants

A third component, the adjuvant, is utilized to boost the immune response of cancer vaccines. An adjuvant is non-immunogenic, but its presence on a microscale significantly improves antibody production and cell-mediated immune response when administered together with the antigen-carrier conjugate vaccines [15]. The most popular molecular immunological adjuvants used in anti-tumor vaccines are QS-21A, Montanide ISA 720, Montanide ISA-51, stimulator of interferon gene protein (STING) agonists, Freund’s adjuvant (IFA), the TLR1/2 ligand adjuvant Amplivant, pathogen-associated molecular pattern molecules (PAMPs), TiterMaxGold, and Sigma adjuvant system (SAS) [23,65,66,67]. The structure of QS-21A contains a complex triterpene–oligosaccharide–normonoterpene conjugate: a quillai acid as a central lipophilic core connected with a linear trisaccharide, and an extended glycosylation diester side chain along its periphery [68]. Initially, it was isolated and purified from the extracts of *Quillaja saponaria*, the soapbark tree, which is known to exhibit remarkable adjuvant activity [68]. Total synthesis of QS-21A_api_ was also accomplished by Gin et al. through application of novel glycosylation methodologies [65]. On the other hand, the Montanide ISA 720 and Montanide ISA-51 adjuvants are formulated as water-in-oil vaccine emulsions and show potential immunoadjuvant activity [15]. Montanide ISA 720 is made of natural metabolizable non-mineral oil and a highly refined emulsifier from the mannide mono-oleate family [15]. Usually, it forms a depot at the injection site, allowing it to slowly release the antigen(s), resulting in enhanced cellular and humoral immune responses [15]. Other newer adjuvants are also being investigated to increase the efficacy of cancer vaccines that target specific immunological pathways. There have been many reports on synthetic agonists which show potential adjuvating activities during cancer vaccine construction. For example, Brimble et al. have reported the synthesis and evaluation of novel TLR2 agonists, such as lipopeptides possessing the S-[2,3-bis(palmitoyloxy)propyl]-L-cysteine (Pam_2_Cys) motif that exhibit potential immunostimulatory effects [67]. As TLR2 recognizes these Pam_2_Cys and Pam_3_Cys motifs, they have been considered attractive components of therapeutic vaccine constructs, due to their well-defined structures and synthetic accessibility. [15] Initially, Pam_1_Cys was synthesized from l-cystine in 7 steps following traditional peptide synthesis protocols [69,70]. Later, they showed that the efficiency of the process could be significantly improved using a thiol–ene strategy [69]. Many synthetic approaches have also been proposed by others.

## 3. Synthetic Biology Approaches for Vaccine Construction and Improving the Potency

### 3.1. Cell-Based Cancer Vaccine

The oldest approach for cancer vaccine development was cell-based, often using tumor cells and tumor lysates [13]. Usually, cell-based vaccines utilize either tumor cells or immune cells, such as T-cells and DCs [13]. However, it has been critical for whole tumor cell vaccines to improve their immunogenicity on their own. Isolated live tumor cells show poor immunogenicity, due to the secretion of immunosuppressive factors, which could affect the activity of immune cells [71]. Therefore, specific synthetic biology approaches were often taken to increase the immunogenicity of tumor cells, thereby improving the efficacy of whole tumor cell vaccines. An example of such an approach was induced immunogenic cell death (ICD) of tumor cells, where the death of tumor cells could incite an adaptive immune response [72]. It has been investigated that compared to replication-deficient adenoviruses or oxaliplatin, oncolytic adenoviruses had better inducement of ICD [72,73]. Using interferon genes (STING)-activating nanoparticles to induce the apoptosis of neuroblastoma cells could also increase immunogenicity [74]. In addition, modifying tumor cells could also improve the efficacy of whole tumor-cell based vaccines. Since interleukins (IL-21 and IL-7) are crucial factors that can strengthen T-cell response, their genetic modification could illustrate significant efficacy of whole tumor cell vaccines [13]. The introduction of double-transfecting IL-15, a modulator of natural killer cells (NK) and memory T-cells, and its truncated receptor IL-15Rα into CT26 colon cancer cells resulted in a robust anti-tumor response, as well [75].

Immune cell-based vaccines have also been used to treat cancer, as seen by DC-based vaccine Sipuleucel-T being successfully used to treat prostate cancer, proving the viability of cancer vaccines and generating great excitement in the cancer vaccine field [13,76]. DCs can be cultured (ex vivo) in two ways: through differentiation from peripheral blood monocytes or from hematopoietic stem cells expressing CD34 [77]. Immature DCs need to be stimulated with a cytokine cocktail (IL-1β, IL-6, TNF-α), CD40L and prostaglandin (PEG)-2, or TLR agonists to produce mature monocyte-derived DC (moDC) [16]. In addition, mRNA can be used to mature DCs. For example, Neyns et al. used mRNA encoding proteins CD40L, CD70, and an active TLR (caTLR4), which were electroporated into DCs to form TriMix-DC [78]. However, natural DCs can present antigens more readily because they express higher amounts of MHC molecules [13]. Therefore, to improve the efficacy of modified DC-based vaccines, external antigens derived from tumor lysates or tumor-derived mRNA, specific TAAs, or TAA-coding mRNA can be introduced [13].

Recently, the concept of chimeric antigen receptor (CAR)-T cell therapy, which can be introduced to improve the efficacy of cell-based vaccines, has gained substantial attention among cancer researchers and oncologists. This therapy is able to be customized by collecting T-cells from a patient via leukapheresis and genetic engineering in the laboratory to express new proteins on their surface, especially CARs [79]. These newly biosynthesized receptors can recognize and bind to specific antigens or epitopes on the surface of cancer cells. Following this procedure, the genetically modified T-cells are allowed to expand in large numbers, then infused back to the patient, where the generated CARs can recognize and attack the tumor cells containing those target antigens on their surface [80]. In addition, DC-based vaccines could be used in combination with adjuvants, cytokines, immune checkpoint inhibitors (ICI), or chemotherapy [13,81].

### 3.2. Virus-Based Vaccines and Virus-like Particles

Many cancer vaccines use either whole (inactivated or live attenuated) microbes or viruses or selected elements from these that are delivered into the body via diverse methods [1,15]. The conversion of life-threatening viruses into vaccines has revolutionized cancer research. Virus-based vaccines prompt the innate and adaptive immune systems to work together to achieve an effective and long-lasting immune response [13]. It has been reported that many forms of cancer are related to viral infections. The most common cancer-related viruses are Epstein–Barr virus (EBV), hepatitis B virus (HBV), hepatitis C virus (HCV), and human papillomavirus (HPV) [82]. Various clinical trials have already proven the anti-tumor efficacy of oncolytic viruses, including herpes simplex virus (HSV), adenovirus, measles, reovirus, and vesicular stomatitis virus [13]. Among these, adenoviruses can be easily manipulated to achieve gene transfer and TA expression [83]. Other vectors like lentiviruses, adeno-associated viruses (AAVs), and vaccinia virus (VACV) are also used on the tumor vaccine platform, in which lentiviruses and adeno-associated viruses have the unique ability to induce stable and long-term expression of the transgene (targeted antigen) in non-dividing cells [13]. These vectors are often utilized in CAR-T therapy, where the T-cells are activated via gene transfer by using retroviral or lentiviral vectors to express transgenes encoding tumor-specific CARs [84]. Additionally, these viral vectors can guide RNA to reverse-transcribe into DNA and permanently integrate into the genome of patient cells [85]. Still, the most effective way to prevent cervical cancer currently is early vaccination against HPV [82]. Cervarix^®^, Gardasil 9^®^, and multivalent Gardasil are the only licensed preventive virus-like particle (VLP)-based HPV vaccines available [82].

However, inactivated whole virus vaccines are being used less frequently in oncological treatments, due to difficulties in production and safety issues [1]. Instead, with the development of bioengineering technology and approaches, VLPs are increasingly being used in vaccine construction [82]. VLPs are self-assembling bio-nanoparticles, similar to antigens, which can present multiple epitopes on their surface in a highly organized manner [1]. Furthermore, VLPs do not contain any viral genetic material, they are non-infectious, and provide significant advantages over live or attenuated viruses [1]. When antigenic determinants are linked to or are associated with the VLP, they can produce a more robust response than traditional carrier proteins [1]. Overall, VLPs can be classified into two categories: non-enveloped and enveloped [82]. For the former, the viral structural protein is fused with a foreign antigen via genetic engineering, followed by the expression of a chimeric protein in a suitable host system (e.g., mammalian cells, insect cells, yeasts, bacteria, or cell-free systems) without obtaining any host component [82]. Furthermore, non-enveloped VLPs can also be chemically conjugated with a targeted antigen via a bifunctional chemical linker, such as sulfosuccinimidyl 4-(N-maleimidomethyl)cyclohexane-1-carboxylate (sulfo-SMCC) and nanoglue, in which, chimeric VLPs can be produced without extensive genetic alteration, overcoming the limitation imposed on VLP formation [82]. In contrast, enveloped VLPs acquire part of the host cell membranes as their lipid envelope, where an epitope could be integrated and displayed on the surface [82]. Alternatively, a protein transfer technique can be utilized to display the heterologous epitope on the surface of enveloped VLPs [82]. This process allows the incorporation of the glycosylphosphatidylinositol (GPI)-anchored proteins or other immunostimulatory molecules to the lipid bilayer of the enveloped VLPs via a simple incubation step [86]. This kind of genetic fusion of a TA to VLPs was previously reported to notably improve antigen delivery and immunogenicity [82].

### 3.3. Nucleic Acid-Based Vaccines

Nucleic acid vaccines (NAVs) have recently been considered a promising cancer therapy. The advantages of nucleic acid synthesis has allowed many synthetic biologists to reengineer entire viral genomes using identical large-scale mutations [12]. The basic principle of NAVs is based on incorporating DNA or RNA encoding viral components into human cells, so that these components can replicate viral antigenic peptides and induce robust cellular and humoral immunity during the relapse of any disease, including cancer [87]. DNA and mRNA are different structurally, with DNA having a double-stranded structure, deoxythymidine, a C2′-endo conformation, and 2′-H in deoxyribose, whereas mRNA has a single-stranded structure, the presence of uridine, a C3′-endo conformation, and 2′-OH in ribose [87]. DNA and mRNA cancer based-vaccines generally deliver genetic information encoding TAs to the host, producing immune responses against TA-expressing cancer cells [87]. Although synthetic DNA-based or RNA-based vaccines are simpler to manufacture and avoid using complex live attenuated viruses, they have not so far been considered viable alternatives to TACA-based vaccines [1,87]. For DNA vaccine construction, plasmid DNA structures are most often used. [88]. In general, the manufacturing of DNA vaccines includes bacterial amplification of the recombinant plasmid, plasmid isolation and purification, transfection into the target cells, transcription, and translation to generate TAs within the target cells [87,88]. The production of plasmid DNA vaccines has some drawbacks, including the time-consuming process and the need to remove impurities from bacterial cultures [87]. On the other hand, RNA vaccines traditionally consist of mRNA synthesized by in vitro transcription (IVT) using a bacteriophage RNA polymerase and template DNA that encodes the desired antigens [89]. These chemically synthesized mRNA vaccines are cell-free and more straightforward than the DNA vaccines, as they use linear DNA molecules or libraries of cDNA for IVT, followed by purification [89].

Nowadays, a polymerase chain reaction (PCR) approach is being used to amplify the desired DNA sample during vaccine construction [87]. Modern DNA vaccines have increased immunogenicity via codon optimization, the co-administration of cytokines, streamlined plasmids, plasmid-free double-stranded DNA (dsDNA) designs, and vaccine delivery through electroporation [87,90]. A new generation of DNA vaccines was constructed by Vandermeulen et al., who encoded a bio-engineered vesicular stomatitis virus glycoprotein as a carrier of T cell tumor epitopes (plasmid to deliver T cell epitopes, pTOP) to improve the stability of naked IVT mRNA-based vaccines, while also applying many encapsulating agents, such as cationic liposomes and *N*-[1-(2,3-dioleoloxy)propyl]-*N*, *N*,*N*-trimethyl ammonium chloride 1(DOTAP), to enhance immune response [88]. Until now, many synthetic biology approaches have been investigated to increase the intracellular stability of RNA vaccines, reduce toxicity, and enhance protein production [87]. For example, the structural alteration of RNAs by addition of complete 5′ Cap1 (N7MeGpppN2′-OMe) during IVT has been shown to enhance mRNA stability by avoiding innate immune recognition of uncapped 5′-triphosphate RNAs [12]. Another approach to increase protein production from mRNA vaccines has been through the use of synthetic self-amplifying mRNAs (saRNAs), which can be made using parts of viruses, such as Semliki Forest virus and Sindbis virus [12,89]. Reinhard et al. have introduced the protein claudin 6 (CLDN6) as a new CAR-T cell target and constructed a RNA vaccine encoding a CAR directed towards CLDN6. This RNA vaccine can also promote CLDN6 expression on the surface of DC cells, thereby improving tumor therapies [91].

### 3.4. Bacteria and Plant-Based Vaccine

Many studies have shown that some specific bacteria can migrate to hypoxic areas of the tumor microenvironment (TME), stimulating an anti-tumor immune reaction, thus becoming a promising platform for cancer treatment [92,93]. In 1868, William Coley infected cancer patients with *Streptococcus pyogenes* species of bacteria. [94] Surprisingly, some patients witnessed reduced tumor size or complete disappearance of tumors, suggesting that bacterial therapy could be a valuable option for cancer research [94]. The only licensed example of a live bacteria-based vaccine being used to treat cancer is the Bacillus Calmette-Guérin (BCG) vaccine, which contains an attenuated strain of *Mycobacterium bovis*, and has been successfully applied to treat superficial bladder cancer [95]. The use of live bacteria as attenuated bacterial vectors for cancer vaccine development have been widely reported in several pre-clinical trials. For example, the cancer-specific antigen human papillomavirus type 16 E7 (HPV-16E7) was fused with an attenuated sequence of Listeriolysin O (LLO), a hemolysin protein produced by *Listeria monocytogenes*, to develop a therapeutic cervical cancer vaccine [96]. Furthermore, an oral DNA vaccine (VXM01) used a licensed live and attenuated *Salmonella* ser. Typhimurium strain Ty21a as a vector [97]. Furthermore, the safety and efficacy of the recombinant *Lactococcus lactis* expressing the HPV type 16 E7 oncogene as a preventive vaccine was evaluated on healthy female volunteers [98].

However, the idea of applying bacteria directly was questioned for a long time, due to the risk of fatal infections. This gave rise to chemically modified and bio-engineered bacteria being considered safer for clinical applications. For instance, positively charged nanoparticles can be self-assembled to the negatively charged cell wall of bacterial species, such as *Salmonella*, through electrostatic interaction. Inspired by this concept, Hu et al. designed a cationic nanoparticle-coated bacteria assembled with a cationic polymer and plasmid DNA to synthesize a nanoparticle-coated attenuated bacteria for an oral DNA-based tumor vaccine [99]. Furthermore, a nonpathogenic *Escherichia coli* strain was bio-engineered with a controllable nanobody (anti-CD47) to effectively activate the infiltration of cytotoxic T lymphocytes [100]. Bacteria can also be genetically modified to express TAAs, bacterial toxins, and cytokines [94]. For instance, *Salmonella* ser. Typhimurium was engineered to express bacterial toxins, such as colicin E3 (ColE3) or HlyE (ClyA), which have also successfully demonstrated antitumor abilities [101]. In fact, many proteins have been used to modify bacterial vaccines, including proteins from the TNF-α family, HPV16-E7, VEGFR2, NY-ESO-1 endoglin, etc. [14]. In addition, antitumor efficacy was amplified when these proteins were fused with a truncated version of LLO or cholera toxin subunit B from *Vibrio cholerae* [14].

On the other hand, plant-derived TAA proteins also hold promise for novel cancer vaccine development. Currently, several plant-based cancer vaccines are under investigation. Similar to nucleic acid-based vaccines, plant-based vaccines mostly rely on the transfection of specific DNA sequences from targeted antigens (transgenes) into the vector [102,103]. The transgene is then expressed in the plant cells using a transient expressing system [102]. Significant advantages of plant-based vaccines over other recombinant vaccines include the low cost, easy scale-up, high biosynthetic production capacity, and lack of these being human pathogens, leading to better clinical safety [104]. Additionally, since the rigid cell walls from plants can protect antigens from degradation under the stomach’s acidic environment, they therefore, can be considered for oral vaccine development [103].

## 4. Biomaterial Scaffolds as Vaccine Delivery Systems

An appropriate design for the delivery of any vaccine and adjuvants can ensure the cytosolic entry and the release of antigenic epitopes, thereby effectively initiating the cross-presentation of antigens to T cells [15]. It is essential to ensure the colocalization and retention time of the adjuvant and antigen in the lymphatic system, where the signals to immune cells are magnified [13]. Furthermore, if the vaccine components stay dispersed in systemic circulation, the potency of that vaccine is significantly reduced and can cause adverse responses, especially inflammation [15]. Current nanomaterial-based strategies for recruiting and activating DCs for cancer vaccines have achieved promising results, but many issues related to safety and effectiveness remain unresolved [5]. More straightforward and safer delivery systems are thus, urgently required to overcome tumors. The most commonly used systems include liposomes, polymeric nanoparticles, self-assembled nanoparticles, lipid nanoparticles, micelles, carbon nanotubes, gold nanoparticles, and virus nanoparticles (Figure 5), that which can be used alone or in combination [5]. Among these, liposomes are popular, as they are versatile and can be modified in various properties by changing lipid composition, charge, and surface properties. However, some disadvantages related to liposomes are low loading capacity, low stability, and toxicity [5]. A novel antigen delivery system based on polysaccharide-coated gold nanoparticles was developed by Barchi et al., targeting APCs expressing dectin-1 [105]. The synthesis of polysaccharide-coated gold nanoparticles was initiated with the dissolution of yeast-derived *β*-1,3-glucans (B_13_G) in NaOH solution, followed by the addition of HAuCl_4_ and heating under microwave irradiation, yielding highly uniform and serum-stable particles [105]. On the other hand, self-assembling peptides can be used to deliver epitopes, antigens, and adjuvants to target cells [19]. Supramolecular self-assembling peptides have a wide range of properties and can be made from nanofibers, nanotubes, nanoribbons, nano-micelles, nanovesicles, and hydrogels [19]. Recently, a tumor-penetrating peptide Fmoc-KCRGDK-based hydrogel formulation was used in a personalized cancer vaccine (PVAX) by Wang et al., encapsulating a bromodomain-containing protein 4 (BRD4) inhibitor JQ1, autologous tumor cells, and a photothermal agent (indocyanine green) [19]. Laser irradiation of PVAX released tumor-specific antigens and significantly inhibited tumor relapse by magnifying the maturation of DC and cytotoxic lymphocytes [19].

Multiple immunosuppressive features of the TME have been shown to affect the delivery of therapeutic cancer vaccines, thereby reducing efficacy and potency. Therefore, many ongoing studies and clinical trials are targeting the immunosuppressive TME to eradicate tumor cells [92]. Recently, injectable hydrogels have gained considerable attraction as vehicles for cancer vaccine delivery in situ because of their properties, especially their unique porous structures that can swell in water or biological liquids [5]. Controlled vaccine release by injectable hydrogels can activate the systematic anti-tumor immune response while causing minimal toxicity [106]. Wang et al. have also designed an injectable PEG-*b*-poly(L-alanine) hydrogel to co-deliver a tumor vaccine and dual ICI to increase immunotherapeutic efficacy [107]. Tumor cell lysates, granulocyte-macrophage colony-stimulating factor (GM-CSF), and ICI were readily encapsulated in the porous hydrogel during spontaneous self-assembly of the polypeptide in an aqueous solution [107]. In addition, a vaccine nodule was developed by Yang et al., consisting of RADA_16_ peptide nanofibrous hydrogel, anti-PD-1 antibodies, tumor cell lysates, and DCs. In both cases, antigen release and improved immune responses were observed [20].

## 5. Outlook and Perspectives

Tumor cells can escape the immune system, which can downregulate TA expression, resulting in poor therapeutic effects of vaccine candidates. Although many cancer vaccines have been reported to be highly immunogenic, most of them have failed in different stages of clinical trials. Therefore, the development of cancer vaccines is very challenging and requires a calculated design plan that focuses on reversing the immunosuppressive effect of TME. In this review, we have highlighted the potential of synthetic chemical and biological approaches for the development of novel cancer vaccines. These chemical strategies are advantageous, as researchers can synthesize novel TAs or epitopes, adjuvants, and biomaterial scaffolds for effective immune responses. Additionally, synthetic biology techniques can be applied for the development of genetically modified tumor vaccines derived from pathogens or other organisms. Although a considerable number of cancer vaccines are based on TAs, nucleic acids, and tumor cells that have been synthesized and processed for clinical trials, the overall efficacy has remained low. Recently, ICI drugs have been getting much attention and exhibit complete eradication of tumors, as well as promising survival rates. Therefore, the immune responses of existing vaccines with adjuvants and ICI drugs loaded in biodegradable materials should be evaluated. The development of modified receptors to analyze the binding motif of desired antigens, thereby predicting the clinical outcome of a vaccine is also necessary. Moreover, the collaboration between different fields is essential to direct future innovations in this area, accelerate the discovery of novel tumor vaccines, and benefit cancer patients, present and future.

## Figures and Tables

**Figure 1 molecules-27-06933-f001:**
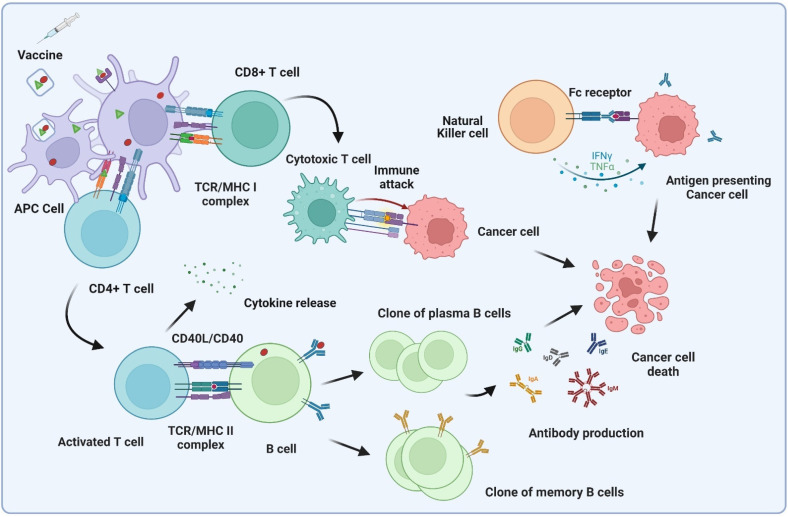
Brief illustration of the antigen-mediated immune response toward cancer cells. Abbreviations: APC, antigen-presenting cell; TCR, T-cell receptor; MHC, major histocompatibility complex.

**Figure 2 molecules-27-06933-f002:**
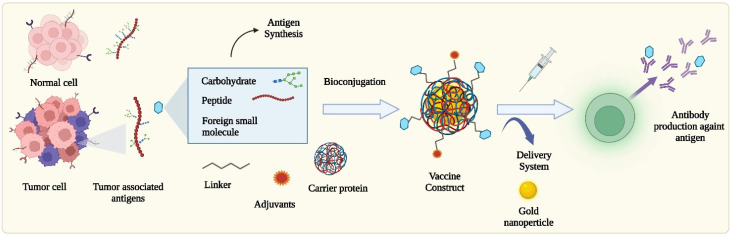
Strategies for tumor antigen-based cancer vaccine development. TAs are overexpressed on the surface of tumor cells due to mutations. These antigens can be synthesized or bioconjugated with carrier proteins by using specific linkers to construct the vaccines. Adjuvants and various delivery systems have been employed to boost the immunogenicity and deliver the vaccines more effectively. Cancer vaccines motivate the production of antibodies against specific TAs on tumor cells.

**Figure 3 molecules-27-06933-f003:**
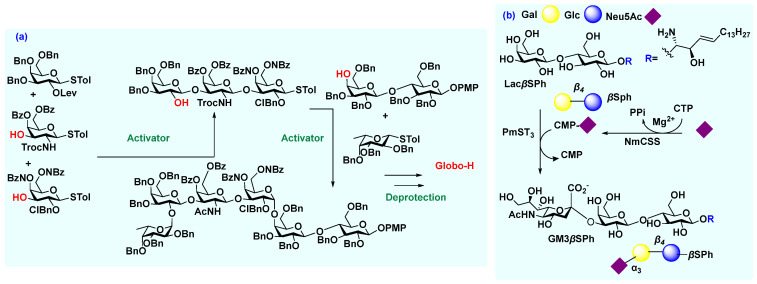
Synthesis of tumor antigens: (**a**) synthesis of Globo-H via glycosylation; (**b**) chemoenzymatic synthesis of GM3*β*SPh. Abbreviation: see Section 2.1.2 & Section 2.1.2.

**Figure 4 molecules-27-06933-f004:**
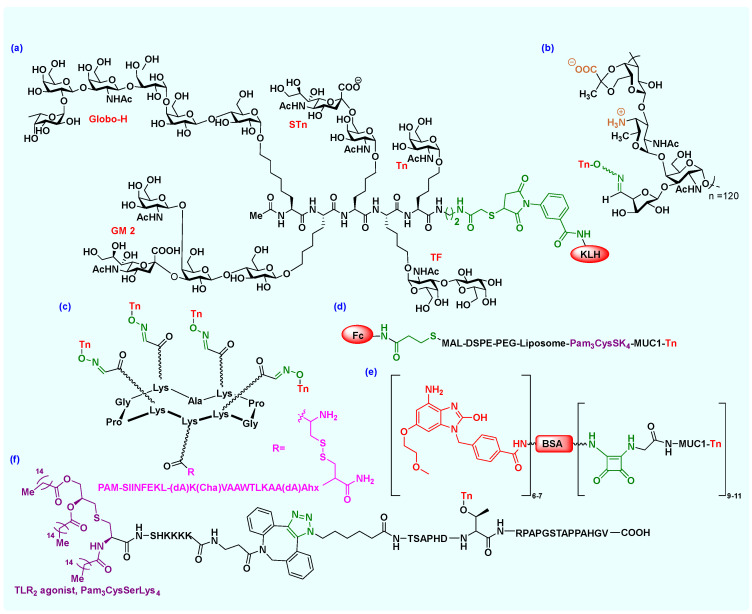
Fully synthetic or semi-synthetic cancer vaccine constructs: (**a**) a multicomponent vaccine containing different TACAs, a peptide backbone, and a carrier protein (KLH); (**b**) Tn antigen conjugated with PS A1 vaccine without any linker; (**c**) a cyclo-decapeptide platform containing clustered Tn antigen analogs, MHC-II epitope, and TLR2 adjuvant; (**d**) liposomal Fc domain conjugated to a MUC1 based cancer vaccine; (**e**) synthesis of TLR7a-BSA-MUC1 vaccine using squaric acid monoamide linker; (**f**) utilization of click chemistry for the synthesis of Pam_3_CysSK_4_-DBCO-MUC1 VNTR-TACA. Abbreviation: dA, D-alanine; Cha, cyclohexyl alanine; Ahx, L-2-aminohexanoic acid; PAM, palmitic acid; Pam3Cys, S-((R)-2,3-bis(palmitoyloxy)propyl)-N-palmitoyl-L-cysteine.

**Figure 5 molecules-27-06933-f005:**
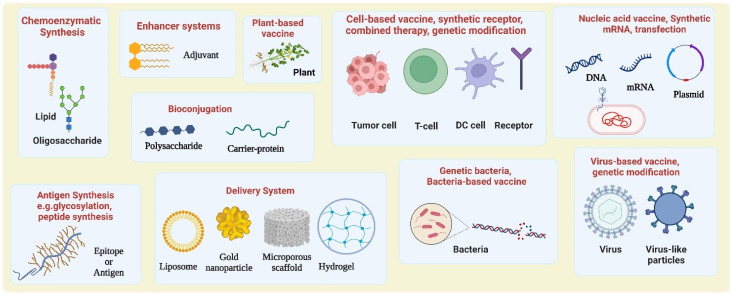
A brief summary of current chemical and synthetic biology approaches for developing cancer vaccines. Abbreviations: DC, dendritic cell.

**Table 1 molecules-27-06933-t001:** Representative clinical trials utilizing antigens or biological materials in cancer vaccines.

NCT Number (Status)	Antigens or Biological	Cancer Type	Clinical Phase
NCT00003871 (Completed)	fowlpox virus, recombinant vaccinia prostate-specific antigen	Prostate cancer	II
NCT00128661 (Completed)	human papillomavirus 16/18 L1 virus-like particle/AS04, hepatitis A inactivated virus vaccine	Cervical cancer (pre-cancerous)	III
NCT01031719 (Completed)	adjuvanted A (H1N1) influenza vaccine, non-adjuvanted A (H1N1) influenza vaccine	Invasive solid tumors	III
NCT01263327 (Completed)	HPV 16/18	Cervical cancer	I
NCT00579423 (Completed)	Globo H, Lewisy and GM2, and glycosylated MUC-1, Tn, and TF	Prostate cancer	II
NCT01248273 (Completed)	Globo-H-GM2-sTn-TF-Tn-KLH conjugate & QS-21	Ovarian cancer, Peritoneal cancer	I
NCT01141491 (Completed)	GM2, GD2, GD3 & OPT-821	Sarcoma	II
NCT00911560 (Active, not recruiting)	GD2L and GD3L conjugated to KLH & OPT-821. Oral β-glucan	Neuroblastoma	I/II
NCT00854789 (Completed)	E75+GM-CSF	Breast cancer	I
NCT00892567 (Completed)	Her-2/neu, CEA& CTA	Breast cancer	I
NCT05013216 (Recruiting)	KRAS Peptide	Pancreatic cancer	I
NCT02019524 (Completed)	E39 & J65 peptide	Breast cancer, Ovarian cancer	I
NCT04024800 (Active, not recruiting)	AE37 peptide, Pembrolizumab	Triple-negative breast cancer	II
NCT01789099 (Active, not recruiting)	UV1 synthetic peptide vaccine, GM-CSF	Non-small cell lung cancer	I/II
NCT04270149 (Recruiting)	ESR1 peptide vaccine	Breast cancer	I
NCT05479045 (Not yet recruiting)	NY-ESO-1 peptide vaccine	Ovarian cancer	II
NCT00681252 (Completed)	URLC10, VEGFR1, VEGFR2	Gastric cancer	I/II
NCT00681330 (Completed)	URC10, TTK, KOC1	Esophageal cancer	I/II
NCT00655785 (Completed)	VEGFR1-1084, VEGFR2-169	Pancreatic cancer	I/II
NCT00433745 (Completed)	WT1 peptide vaccine	Acute myeloid leukemia (AML), Chronic myeloid leukemia (CML)	II
NCT01232712 (Completed)	ImMucin, hGM-CSF	Multiple myeloma	I/II
NCT00254397 (Completed)	GP100: 209-217(210M), MAGE-3 peptide	Melanoma	II
NCT00499577 (Completed)	CMV pp65, hTERT I540/R572Y/D988Y multi-peptide, pneumococcal polyvalent vaccine, survivin (Sur1M2) vaccine	Multiple myeloma, Plasma cell neoplasm	I/II
NCT00019929 (Completed)	mutant p53 peptide-pulsed dendritic cell vaccine	Lung cancer	II
NCT00299728 (Completed)	NY-ESO-1 protein	Various cancers	I
NCT01522820 (Completed)	DEC-205/NY-ESO-1 Fusion protein CDX-1401	Various cancers	I
NCT00705835 (Completed)	rsPSMA protein & Alhydrogel vaccine	Prostate cancer	I
NCT00503568 (Completed)	Ad100-gp96Ig-HLA A1	Lung cancer	I
NCT00072085 (Completed)	Aldesleukin, gp100 antigen, incomplete Freund’s adjuvant	Melanoma	II
NCT00142454 (Completed)	NY-ESO-1 protein & Imiquimod	Melanoma (malignant)	I
NCT04521764 (Recruiting)	Oncolytic Measles Virus encoding Helicobacter pylori neutrophil-activating protein	Breast cancer, stage IV	I
NCT00343109 (Completed)	HER-2/neu	Breast cancer, stages III and IV	II
NCT00204516 (Completed)	mRNA coding for melanoma-associated antigens & GM-CSF	Malignant melanoma	I/II
NCT04847050 (Recruiting)	mRNA-1273	Solid tumor malignancy, Leukemia, Lymphoma, Multiple myeloma	II
NCT03897881 (Active, not recruiting)	mRNA-4157 & Pembrolizumab	Melanoma	II
NCT01995708 (Completed)	CT7, MAGE-A3, and WT1 mRNA-electroporated Langerhans cells	Multiple myeloma	I
NCT04573140 (Recruiting)	Autologous total tumor mRNA and pp65 full length lysosomal associated membrane protein (LAMP), mRNA loaded DOTAP liposome vaccine administered intravenously (RNA loaded lipid particles, RNA-LPs)	Glioblastoma	I
NCT03688178 (Recruiting)	Human CMV pp65-LAMP mRNA-pulsed autologous DCs	Glioblastoma	II
NCT00807781 (Completed)	Mammaglobin-A DNA vaccine	Metastatic breast cancer	I
NCT02348320 (Completed)	Personalized polyepitope DNA vaccine	Breast cancer	I
NCT00859729 (Completed)	pVAXrcPSAv53l (DNA encoding rhesus PSA)	Prostate cancer	I/II
NCT01706458 (Completed)	Sipuleucel-T	Prostate cancer	II
NCT02018458 (Completed)	LA TNBC; ER+/HER2-BC	Breast cancer	I/II
NCT04348747 (Recruiting)	Anti-HER2/HER3 DC vaccine, Pembrolizumab	Breast cancer, stage IV	II
NCT02061423 (Active, not recruiting)	HER-2 pulsed DC vaccine	Breast cancer	I
NCT01730118 (Completed)	AdHER2/neu DC vaccine	Breast cancer, Adenocarcinomas, Metastatic solid tumors	I

Antigens or biological material is as follows: viruses (in yellow), carbohydrates (in red), peptides (in purple), proteins (in blue), mRNA (in orange), DNA (in pink), and dendritic cells (in green). All information is from ClinicalTrials.gov and Refs. [13,17,18].

## Data Availability

Not applicable.

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
