# Peer review of "Chemical and Synthetic Biology Approaches for Cancer Vaccine Development"

_molecules, 2022, doi:10.3390/molecules27206933_

Round 1

Reviewer 1 Report

This manuscript reviewed and highlighted chemical and synthetic biology approaches to overcome the use of whole organisms or pathogens as cancer vaccines. 

Overall, this review introduced many related studies but failed to deliver a take-home message to readers. It was hard to follow due to a lack of rationale and poor structure. More importantly, this review missed significant points on what are the main huddle in the development of natural cancer vaccines, and how the synthetic approach can overcome those. These need to be addressed in this review. Based on this, the reviewer does not recommend this review for publication in this journal. 

Additional comments need to be addressed:

Line 64-66: The author stated “ This review will highlight the different bioconjugation…”, but the reviewer was not able to see the effort. Small parts in sections 2.2.1 and 2.3.2.

Figure 2: Not clear

Table 1: All are sorted from one source. The readers can see these from the source directly. Is any other additional source available?

Line 110: “only a few cancer vaccines”, then what are they? 

Figure 3: Not clear but confusing. All abbreviations should be defined. PAM and Pam are the same ones? dA is not L-alanine, but D-Alanine. The reviewer suggests the authors add statements about structure design and strategy for readers.

Lines 312-347: Peptide synthesis is a well-known method, and there is no specific information on applying this tool for cancer vaccine development. Add specific statements or delete them.

Line 345: PyBO should be PyBOP

Author Response

Reviewer 1 (Round 1)

This manuscript reviewed and highlighted chemical and synthetic biology approaches to overcome the use of whole organisms or pathogens as cancer vaccines. 

Overall, this review introduced many related studies but failed to deliver a take-home message to readers. It was hard to follow due to a lack of rationale and poor structure. More importantly, this review missed significant points on what are the main huddle in the development of natural cancer vaccines, and how the synthetic approach can overcome those. These need to be addressed in this review. Based on this, the reviewer does not recommend this review for publication in this journal. 

Ans: The aim of this review is to give a comprehensive summary of recent advances in cancer vaccine development and the focus is specifically on chemical and synthetic biology approaches.

Additional comments need to be addressed:

  1. Line 64-66: The author stated “ This review will highlight the different bioconjugation…”, but the reviewer was not able to see the effort. Small parts in sections 2.2.1 and 2.3.2.

Ans: The authors would like to appreciate the reviewer’s suggestion. However, these sections have already contained several bioconjugation strategies, including detailed examples. We also discussed click chemistry in Section 2.3.3, which can also be utilized as a conjugation strategy in vaccine construction.

  1. Figure 2: Not clear

Ans: The Figure 2 is revised to contain a brief explanation now.  

  1. Table 1: All are sorted from one source. The readers can see these from the source directly. Is any other additional source available?

Ans: More references are included now. Please see line 113.

  1. Line 110: “only a few cancer vaccines”, then what are they? 

Ans: Examples of FDA-approved cancer vaccines are mentioned now in the introduction section. In addition, Section 3.2 also mentioned some approved vaccines.

  1. Figure 3: Not clear but confusing. All abbreviations should be defined. PAM and Pam are the same ones? dA is not L-alanine, but D-Alanine. The reviewer suggests the authors add statements about structure design and strategy for readers.

Ans: All the abbreviations are included and corrected according to the reviewer’s suggestions.

  1. Lines 312-347: Peptide synthesis is a well-known method, and there is no specific information on applying this tool for cancer vaccine development. Add specific statements or delete them.

Ans: Peptide synthesis is found to be an important protocol for vaccine synthesis because many cancer vaccines contain peptide epitopes. These epitopes are usually synthesized by using peptide synthesis protocols followed by bioconjugation. In the revised manuscript, clearer statements have been made and corresponding examples have been provided accordingly.

  1. Line 345: PyBO should be PyBOP

Ans: This typo has been revised.  

Reviewer 2 Report

In this review Hossain et al collect the main chemical and synthetic approaches for cancer vaccine development.

The authors have structured the article correctly from a detailed chemical approaches to the synthetic biology approaches and finishing with a short paragraph describing the use of biological scaffolds.

I would highlight the complete Table 1 with representative clinical trials using biological materials in cancer vaccines.

The article in general seems clear to me, with indisputable scientific bibliography.

I would just like to mention some details that could be improve reading and comprehension of the paper.

Minor points:

-          1. Line 59, abbreviation of TACA should be described.

-          2. Same for abbreviation of ZPs, line 250.

-         3.  Same for ICI, line 657.

-         4.  Figure 3, the sections a,b, c,d….are not well visible. I propose to change its positions at the top left of each scheme.

-          A figure or scheme for point 2.1.1 (glycosylation assembly) and 2.1.2 could be useful. These points are explained in great detail and I have doubts about its contribution to the review message.  

-         5.  Figure 4 it is not clear. In the central circle the word “approaches” is missing. The pie segment referring “synthesis” does not agree with the rest of the scheme. Adjuvants could be named as “enhancer systems”. In brief, this figure should be improved.

-        6.   I have missed some reference to inmmunotherapy strategies, i.e,  CARs therapy. This point could be included on cell-based cancer vaccines paragraph (point 3.1) or on virus-based vaccines paragraph (point 3.2) as CARs are usually based on retro or lentiviral transfections.

-         7.  Another subject to consider is the advance in the development of personalized neoantigen-based therapeutic cancer vaccines. A mention on this point could update the review.

Author Response

Reviewer 2 (Round 1)

In this review Hossain et al collect the main chemical and synthetic approaches for cancer vaccine development.

The authors have structured the article correctly from a detailed chemical approaches to the synthetic biology approaches and finishing with a short paragraph describing the use of biological scaffolds.

I would highlight the complete Table 1 with representative clinical trials using biological materials in cancer vaccines.

The article in general seems clear to me, with indisputable scientific bibliography.

I would just like to mention some details that could be improve reading and comprehension of the paper.

Minor points:

  1.  Line 59, abbreviation of TACA should be described.

Ans: We would like to thank the reviewer for pointing it out. The abbreviation of TACA is now included.

  1. Same for abbreviation of ZPs, line 250.

Ans: In line 66, zwitterionic polysaccharides (ZPs) was abbreviated.

  1. Same for ICI, line 657.

Ans: In line 488, immune checkpoint inhibitors (ICI) was abbreviated.  

  1. Figure 3, the sections a,b, c,d….are not well visible. I propose to change its positions at the top left of each scheme.

Ans: The Figure 3 (current Figure 4) is corrected according to the suggestions of both reviewers.  

  1. A figure or scheme for point 2.1.1 (glycosylation assembly) and 2.1.2 could be useful. These points are explained in great detail and I have doubts about its contribution to the review message

Ans:   Figure 3 now indicates the utilization of glycosylation and chemoenzymatic synthesis methods for the synthesis of tumor antigens.

  1. Figure 4 it is not clear. In the central circle the word “approaches” is missing. The pie segment referring “synthesis” does not agree with the rest of the scheme. Adjuvants could be named as “enhancer systems”. In brief, this figure should be improved.

Ans: Figure 4 (current Figure 5) is revised according to the reviewer’s suggestions.

  1.  I have missed some reference to inmmunotherapy strategies, i.e,  CARs therapy. This point could be included on cell-based cancer vaccines paragraph (point 3.1) or on virus-based vaccines paragraph (point 3.2) as CARs are usually based on retro or lentiviral transfections.

Ans: We appreciate the reviewer’s suggestion and included the CAT-therapy concept in Sections 3.1, 3.2, and 3.3. More references regarding this concept are available now.  

  1. Another subject to consider is the advance in the development of personalized neoantigen-based therapeutic cancer vaccines. A mention on this point could update the review.

Ans: We would like to thank the reviewer for the valuable suggestion. We have mentioned neo-antigens, neo-epitopes, and personalized neo-antigen-based vaccines in revised Sections 1 and 2.3.